# Scalable Adaptive Stochastic Optimization Using Random Projections

**Gabriel Krummenacher**♦*
gabriel.krummenacher@inf.ethz.ch

**Brian McWilliams**♥*
brian@disneyresearch.com

**Yannic Kilcher**♦
yannic.kilcher@inf.ethz.ch

**Joachim M. Buhmann**♦
jbuhmann@inf.ethz.ch

**Nicolai Meinshausen**♣
meinshausen@stat.math.ethz.ch

♦Institute for Machine Learning, Department of Computer Science, ETH Zürich, Switzerland
♣Seminar for Statistics, Department of Mathematics, ETH Zürich, Switzerland
♥Disney Research, Zürich, Switzerland

## Abstract

Adaptive stochastic gradient methods such as ADAGRAD have gained popularity in particular for training deep neural networks. The most commonly used and studied variant maintains a diagonal matrix approximation to second order information by accumulating past gradients which are used to tune the step size adaptively. In certain situations the full-matrix variant of ADAGRAD is expected to attain better performance, however in high dimensions it is computationally impractical. We present ADA-LR and RADAGRAD two computationally efficient approximations to full-matrix ADAGRAD based on randomized dimensionality reduction. They are able to capture dependencies between features and achieve similar performance to full-matrix ADAGRAD but at a much smaller computational cost. We show that the regret of ADA-LR is close to the regret of full-matrix ADAGRAD which can have an up-to exponentially smaller dependence on the dimension than the diagonal variant. Empirically, we show that ADA-LR and RADAGRAD perform similarly to full-matrix ADAGRAD. On the task of training convolutional neural networks as well as recurrent neural networks, RADAGRAD achieves faster convergence than diagonal ADAGRAD.

## 1   Introduction

Recently, so-called *adaptive* stochastic optimization algorithms have gained popularity for large-scale convex and non-convex optimization problems. Among these, ADAGRAD [9] and its variants [21] have received particular attention and have proven among the most successful algorithms for training deep networks. Although these problems are inherently highly non-convex, recent work has begun to explain the success of such algorithms [3].

ADAGRAD adaptively sets the learning rate for each dimension by means of a time-varying proximal regularizer. The most commonly studied and utilised version considers only a diagonal matrix proximal term. As such it incurs almost no additional computational cost over standard stochastic

---

gradient descent (SGD). However, when the data has low *effective rank* the regret of ADAGRAD may have a much worse dependence on the dimensionality of the problem than its full-matrix variant (which we refer to as ADA-FULL). Such settings are common in high dimensional data where there are many correlations between features and can also be observed in the convolutional layers of neural networks. The computational cost of ADA-FULL is substantially higher than that of ADAGRAD– it requires computing the inverse square root of the matrix of gradient outer products to evaluate the proximal term which grows with the cube of the dimension. As such it is rarely used in practise.

In this work we propose two methods that approximate the proximal term used in ADA-FULL drastically reducing computational and storage complexity with little adverse affect on optimization performance. First, in Section 3.1 we develop ADA-LR, a simple approximation using random projections. This procedure reduces the computational complexity of ADA-FULL by a factor of $p$ but retains similar theoretical guarantees. In Section 3.2 we systematically profile the most computationally expensive parts of ADA-LR and introduce further randomized approximations resulting in a truly scalable algorithm, RADAGRAD. In Section 3.3 we outline a simple modification to RADAGRAD– reducing the variance of the stochastic gradients – which greatly improves practical performance. Finally we perform an extensive comparison between the performance of RADAGRAD with several widely used optimization algorithms on a variety of deep learning tasks. For image recognition with convolutional networks and language modeling with recurrent neural networks we find that RADAGRAD and in particular its variance-reduced variant achieves faster convergence.

## 1.1 Related work

Motivated by the problem of training deep neural networks, very recently many new adaptive optimization methods have been proposed. Most computationally efficient among these are first order methods similar in spirit to ADAGRAD, which suggest alternative normalization factors [21, 28, 6]. Several authors propose efficient stochastic variants of classical second order methods such as L-BFGS [5, 20]. Efficient algorithms exist to update the inverse of the Hessian approximation by applying the matrix-inversion lemma or directly updating the Hessian-vector product using the "double-loop" algorithm but these are not applicable to ADAGRAD style algorithms. In the convex setting these methods can show great theoretical and practical benefit over first order methods but have yet to be extensively applied to training deep networks.

On a different note, the growing zoo of *variance reduced* SGD algorithms [19, 7, 18] has shown vastly superior performance to ADAGRAD-style methods for standard empirical risk minimization and convex optimization. Recent work has aimed to move these methods into the non-convex setting [1]. Notably, [22] combine variance reduction with second order methods.

Most similar to RADAGRAD are those which propose factorized approximations of second order information. Several methods focus on the natural gradient method [2] which leverages second order information through the Fisher information matrix. [14] approximate the inverse Fisher matrix using a sparse graphical model. [8] use low-rank approximations whereas [26] propose an efficient Kronecker product based factorization. Concurrently with this work, [12] propose a randomized preconditioner for SGD. However, their approach requires access to all of the data at once in order to compute the preconditioning matrix which is impractical for training deep networks. [23] propose a theoretically motivated algorithm similar to ADA-LR and a faster alternative based on Oja's rule to update the SVD.

**Fast random projections.** Random projections are low-dimensional embeddings $\mathbf{\Pi} : \mathbb{R}^p \to \mathbb{R}^\tau$ which preserve – up to a small distortion – the geometry of a subspace of vectors. We concentrate on the class of *structured* random projections, among which the Subsampled Randomized Fourier Transform (SRFT) has particularly attractive properties [15]. The SRFT consists of a preconditioning step after which $\tau$ columns of the new matrix are subsampled uniformly at random as $\mathbf{\Pi} = \sqrt{p/\tau}\mathbf{S\Theta D}$ with the definitions: **(i)** $\mathbf{S} \in \mathbb{R}^{\tau \times p}$ is a subsampling matrix. **(ii)** $\mathbf{D} \in \mathbb{R}^{p \times p}$ is a diagonal matrix whose entries are drawn independently from $\{-1, 1\}$. **(iii)** $\mathbf{\Theta} \in \mathbb{R}^{p \times p}$ is a unitary discrete Fourier trananform (DFT) matrix. This formulations allows very fast implementations using the fast Fourier transform (FFT), for example using the popular FFTW package[2]. Applying the FFT to a $p-$dimensional vector can be achieved in $O\left(p \log \tau\right)$ time. Similar structured random projections

have gained popularity as a way to speed up [24] and robustify [27] large-scale linear regression and for distributed estimation [17, 16].

## 1.2 Problem setting

The problem considered by [9] is online stochastic optimization where the goal is, at each step, to predict a point $\boldsymbol{\beta}_t \in \mathbb{R}^p$ which achieves low regret with respect to a fixed optimal predictor, $\boldsymbol{\beta}^{\text{opt}}$, for a sequence of (convex) functions $F_t(\boldsymbol{\beta})$. After $T$ rounds, the regret can be defined as $R(T) = \sum_{t=1}^{T} F_t(\boldsymbol{\beta}_t) - \sum_{t=1}^{T} F_t(\boldsymbol{\beta}^{\text{opt}})$.

Initially, we will consider functions $F_t$ of the form $F_t(\boldsymbol{\beta}) := f_t(\boldsymbol{\beta}) + \varphi(\boldsymbol{\beta})$ where $f_t$ and $\varphi$ are convex loss and regularization functions respectively. Throughout, the vector $\mathbf{g}_t \in \nabla f_t(\boldsymbol{\beta}_t)$ refers to a particular subgradient of the loss function. Standard first order methods update $\boldsymbol{\beta}_t$ at each step by moving in the opposite direction of $\mathbf{g}_t$ according to a step-size parameter, $\eta$. The ADAGRAD family of algorithms [9] instead use an *adaptive* learning rate which can be different for each feature. This is controlled using a time-varying proximal term which we briefly review. Defining $\mathbf{G}_t = \sum_{i=1}^{t} \mathbf{g}_i \mathbf{g}_i^\top$ and $\mathbf{H}_t = \delta \mathbf{I}_p + (\mathbf{G}_{t-1} + \mathbf{g}_t \mathbf{g}_t^\top)^{1/2}$, the ADA-FULL proximal term is given by $\psi_t(\boldsymbol{\beta}) = \frac{1}{2} \langle \boldsymbol{\beta}, \mathbf{H}_t \boldsymbol{\beta} \rangle$.

Clearly when $p$ is large, constructing $\mathbf{G}$ and finding its root and inverse *at each iteration* is impractical. In practice, rather than the full outer product matrix, ADAGRAD uses a proximal function consisting of the diagonal of $\mathbf{G}_t$, $\psi_t(\boldsymbol{\beta}) = \frac{1}{2} \langle \boldsymbol{\beta}, (\delta \mathbf{I}_p + \text{diag}(\mathbf{G}_t)^{1/2}) \boldsymbol{\beta} \rangle$. Although the diagonal proximal term is computationally cheaper, it is unable to capture dependencies between coordinates in the gradient terms. Despite this, ADAGRAD has been found to perform very well empirically. One reason for this is modern high-dimensional datasets are typically also very sparse. Under these conditions, coordinates in the gradient are approximately independent.

## 2 Stochastic optimization in high dimensions

ADAGRAD has attractive theoretical and empirical properties and adds essentially no overhead above a standard first order method such as SGD. It begs the question, what we might hope to gain by introducing additional computational complexity. In order to motivate our contribution, we first present an analogue of the discussion in [10] focussing on when data is high-dimensional and dense. We argue that if the data has low-rank (rather than sparse) structure ADA-FULL can effectively adapt to the intrinsic dimensionality. We also show in Section 3.1 that ADA-LR has the same property.

First, we review the theoretical properties of ADAGRAD algorithms, borrowing the $g_{1:T,j}$ notation[9].

**Proposition 1.** ADAGRAD *and* ADA-FULL *achieve the following regret (Corollaries 6 & 11 from [9]) respectively:*

$$R_D(T) \leq 2\|\boldsymbol{\beta}^{opt}\|_\infty \sum_{j=1}^{p} \|g_{1:T,j}\| + \delta\|\boldsymbol{\beta}^{opt}\|_1 , \qquad R_F(T) \leq 2\|\boldsymbol{\beta}^{opt}\| \cdot \text{tr}(\mathbf{G}_T^{1/2}) + \delta\|\boldsymbol{\beta}^{opt}\|. \quad (1)$$

The major difference between $R_D(T)$ and $R_F(T)$ is the inclusion of the final full-matrix and diagonal proximal term, respectively. Under a sparse data generating distribution ADAGRAD achieves an up-to exponential improvement over SGD which is optimal in a minimax sense [10]. While data sparsity is often observed in practise in high-dimensional datasets (particularly web/text data) many other problems are dense. Furthermore, in practise applying ADAGRAD to dense data results in a learning rate which tends to decay too rapidly. It is therefore natural to ask how dense data affects the performance of ADA-FULL.

For illustration, consider when the data points $\mathbf{x}_i$ are sampled i.i.d. from a Gaussian distribution $P_X = \mathcal{N}(\mathbf{0}, \boldsymbol{\Sigma})$. The resulting variable will clearly be dense. A common feature of high dimensional data is low *effective rank* defined for a matrix $\boldsymbol{\Sigma}$ as $r(\boldsymbol{\Sigma}) = \text{tr}(\boldsymbol{\Sigma})/\|\boldsymbol{\Sigma}\| \leq \text{rank}(\boldsymbol{\Sigma}) \leq p$. Low effective rank implies that $r \ll p$ and therefore the eigenvalues of the covariance matrix decay quickly. We will consider distributions parameterised by covariance matrices $\boldsymbol{\Sigma}$ with eigenvalues $\lambda_j(\boldsymbol{\Sigma}) = \lambda_0 j^{-\alpha}$ for $j = 1, \ldots, p$.

Functions of the form $F_t(\boldsymbol{\beta}) = F_t(\boldsymbol{\beta}^\top \mathbf{x}_t)$ have gradients $\|\mathbf{g}_t\| \leq M \|\mathbf{x}_t\|$. For example, the least squares loss $F_t(\boldsymbol{\beta}^\top \mathbf{x}_t) = \frac{1}{2}(y_t - \boldsymbol{\beta}^\top \mathbf{x}_t)^2$ has gradient $\mathbf{g}_t = \mathbf{x}_t(y_t - \mathbf{x}_t^\top \boldsymbol{\beta}_t) = \mathbf{x}_t \varepsilon_t$, such that

$\|\varepsilon_t\| \leq M$. Let us consider the effect of distributions parametrised by $\Sigma$ on the proximal terms of full, and diagonal ADAGRAD. Plugging $X$ into the proximal terms of (1) and taking expectations with respect to $P_X$ we obtain for ADAGRAD and ADA-FULL respectively:

$$\mathbb{E} \sum_{j=1}^{p} \|g_{1:T,j}\| \leq \sum_{j=1}^{p} \sqrt{M^2 \mathbb{E} \sum_{t=1}^{T} x_{t,j}^2} \leq pM\sqrt{T}, \quad \mathbb{E} \operatorname{tr}((\sum_{t=1}^{T} \mathbf{g}_t \mathbf{g}_t^\top)^{1/2}) \leq M\sqrt{T\lambda_0} \sum_{j=1}^{p} j^{-\alpha/2}, \tag{2}$$

where the first inequality is from Jensen and the second is from noticing the sum of $T$ squared Gaussian random variables is a $\chi^2$ random variable. We can consider the effect of fast-decaying spectrum: for $\alpha \geq 2$, $\sum_{j=1}^{p} j^{-\alpha/2} = O(\log p)$ and for $\alpha \in (1,2)$, $\sum_{j=1}^{p} j^{-\alpha/2} = O(p^{1-\alpha/2})$.

When the data (and thus the gradients) are dense, yet have low effective rank, ADA-FULL is able to adapt to this structure. On the contrary, although ADAGRAD is computationally practical, in the worst case it may have exponentially worse dependence on the data dimension ($p$ compared with $\log p$). In fact, the discrepancy between the regret of ADA-FULL and that of ADAGRAD is analogous to the discrepancy between ADAGRAD and SGD for sparse data.

| **Algorithm 1** ADA-LR | **Algorithm 2** RADAGRAD |
|---|---|
| **Input:** $\eta > 0, \delta \geq 0, \tau$ | **Input:** $\eta > 0, \delta \geq 0, \tau$ |
| 1: **for** $t = 1 \ldots T$ **do** | 1: **for** $t = 1 \ldots T$ **do** |
| 2:    *Receive* $\mathbf{g}_t = \nabla f_t(\boldsymbol{\beta}_t)$. | 2:    *Receive* $\mathbf{g}_t = \nabla f_t(\boldsymbol{\beta}_t)$. |
| 3:    $\mathbf{G}_t = \mathbf{G}_{t-1} + \mathbf{g}_t \mathbf{g}_t^\top$ | 3:    *Project*: $\tilde{\mathbf{g}}_t = \mathbf{\Pi} \mathbf{g}_t$ |
| 4:    *Project*: $\tilde{\mathbf{G}}_t = \mathbf{G}_t \mathbf{\Pi}$ | 4:    $\tilde{\mathbf{G}}_t = \tilde{\mathbf{G}}_{t-1} + \mathbf{g}_t \tilde{\mathbf{g}}_t^\top$ |
| 5:    $\mathbf{QR} = \tilde{\mathbf{G}}_t$ {QR-decomposition} | 5:    $\mathbf{Q}_t, \mathbf{R}_t \leftarrow \texttt{qr\_update}(\mathbf{Q}_{t-1}, \mathbf{R}_{t-1}, \mathbf{g}_t, \tilde{\mathbf{g}}_t)$ |
| 6:    $\mathbf{B} = \mathbf{Q}^\top \mathbf{G}_t$ | 6:    $\mathbf{B} = \tilde{\mathbf{G}}_t^\top \mathbf{Q}$ |
| 7:    $\mathbf{U}, \boldsymbol{\Sigma}, \mathbf{V} = \mathbf{B}$ {SVD} | 7:    $\mathbf{U}, \boldsymbol{\Sigma}, \mathbf{W} = \mathbf{B}$ {SVD} |
| 8: | 8:    $\mathbf{V} = \mathbf{W}\mathbf{Q}^\top$ |
| 9: | 9:    $\boldsymbol{\gamma}_t = \eta(\mathbf{g}_t - \mathbf{V}\mathbf{V}^\top \mathbf{g}_t)$ |
| 10:    $\boldsymbol{\beta}_{t+1} = \boldsymbol{\beta}_t - \eta \mathbf{V}(\boldsymbol{\Sigma}^{1/2} + \delta\mathbf{I})^{-1}\mathbf{V}^\top \mathbf{g}_t$ | 10:    $\boldsymbol{\beta}_{t+1} = \boldsymbol{\beta}_t - \eta \mathbf{V}(\boldsymbol{\Sigma}^{1/2} + \delta\mathbf{I})^{-1}\mathbf{V}^\top \mathbf{g}_t - \boldsymbol{\gamma}_t$ |
| 11: **end for** | 11: **end for** |
| **Output:** $\boldsymbol{\beta}_T$ | **Output:** $\boldsymbol{\beta}_T$ |

## 3 Approximating ADA-FULL using random projections

It is clear that in certain regimes, ADA-FULL provides stark *optimization* advantages over ADAGRAD in terms of the dependence on $p$. However, ADA-FULL requires maintaining a $p \times p$ matrix, $\mathbf{G}$ and computing its square root and inverse. Therefore, *computationally* the dependence of ADA-FULL on $p$ scales with the cube which is impractical in high dimensions.

A naïve approach would be to simply reduce the dimensionality of the gradient vector, $\tilde{\mathbf{g}}_t \in \mathbb{R}^\tau = \mathbf{\Pi}\mathbf{g}_t$. ADA-FULL is now directly applicable in this low-dimensional space, returning a solution vector $\tilde{\boldsymbol{\beta}}_t \in \mathbb{R}^\tau$ at each iteration. However, for many problems, the original coordinates may have some intrinsic meaning or in the case of deep networks, may be parameters in a model. In which case it is important to return a solution in the original space. Unfortunately in general it is not possible to recover such a solution from $\tilde{\boldsymbol{\beta}}_t$ [30].

Instead, we consider a different approach to maintaining and updating an approximation of the ADAGRAD matrix while retaining the original dimensionality of the parameter updates $\boldsymbol{\beta}$ and gradients $\mathbf{g}$.

### 3.1 Randomized low-rank approximation

As a first approach we approximate the inverse square root of $\mathbf{G}_t$ using a fast randomized singular value decomposition (SVD) [15]. We proceed in two stages: First we compute an approximate basis

$\mathbf{Q}$ for the range of $\mathbf{G}_t$. Then we use $\mathbf{Q}$ to compute an approximate SVD of $\mathbf{G}_t$ by forming the smaller dimensional matrix $\mathbf{B} = \mathbf{Q}^\top \mathbf{G}_t$ and then compute the low-rank SVD $\mathbf{U\Sigma V}^\top = \mathbf{B}$. This is faster than computing the SVD of $\mathbf{G}_t$ directly if $\mathbf{Q}$ has few columns.

An approximate basis $\mathbf{Q}$ can be computed efficiently by forming the matrix $\tilde{\mathbf{G}}_t = \mathbf{G}_t \mathbf{\Pi}$ by means of a structured random projection and then constructing an orthonormal basis for the range of $\tilde{\mathbf{G}}_t$ by QR-decomposition. The randomized SVD allows us to quickly compute the square root and pseudo-inverse of the proximal term $\mathbf{H}_t$ by setting $\tilde{\mathbf{H}}_t^{-1} = \mathbf{V}(\mathbf{\Sigma}^{1/2} + \delta \mathbf{I})^{-1}\mathbf{V}^\top$. We call this approximation ADA-LR and describe the steps in full in Algorithm 1.

In practice, using a structured random projection such as the SRFT leads to an approximation of the original matrix, $\mathbf{G}_t$ of the following form $\left\| \mathbf{G}_t - \mathbf{Q}\mathbf{Q}^\top \mathbf{G}_t \right\| \leq \epsilon$, with high probability [15] where $\epsilon$ depends on $\tau$, the number of columns of $\mathbf{Q}$; $p$ and the $\tau^{th}$ singular value of $\mathbf{G}_t$. Briefly, if the singular values of $\mathbf{G}_t$ decay quickly and $\tau$ is chosen appropriately, $\epsilon$ will be small (this is stated more formally in Proposition 2). We leverage this result to derive the following regret bound for ADA-LR (see C.1 for proof).

**Proposition 2.** *Let $\sigma_{k+1}$ be the kth largest singular value of $\mathbf{G}_t$. Setting the projection dimension as* $4\left(\sqrt{k} + \sqrt{8\log(kn)}\right)^2 \leq \tau \leq p$ *and defining* $\epsilon = \sqrt{1 + 7p/\tau} \cdot \sigma_{k+1}$. *With failure probability at most* $O\left(k^{-1}\right)$ ADA-LR *achieves regret* $R_{LR}(T) \leq 2\|\boldsymbol{\beta}^{opt}\|\text{tr}(\mathbf{G}_T^{1/2}) + (2\tau\sqrt{\epsilon} + \delta)\|\boldsymbol{\beta}^{opt}\|$ .

Due to the randomized approximation we incur an additional $2\tau\sqrt{\epsilon}\|\boldsymbol{\beta}^{\text{opt}}\|$ compared with the regret of ADA-FULL (eq. 1). So, under the earlier stated assumption of fast decaying eigenvalues we can use an identical argument as in eq. (2) to similarly obtain a dimension dependence of $O(\log p + \tau)$.

Approximating the inverse square root decreases the complexity of each iteration from $O\left(p^3\right)$ to $O\left(\tau p^2\right)$. We summarize the cost of each step in Algorithm 1 and contrast it with the cost of ADA-FULL in Table A.1 in Section A. Even though ADA-LR removes one factor of $p$ form the runtime of ADA-FULL it still needs to store the large matrix $\mathbf{G}_t$. This prevents ADA-LR from being a truly practical algorithm. In the following section we propose a second algorithm which directly stores a low dimensional approximation to $\mathbf{G}_t$ that can be updated cheaply. This allows for an improvement in runtime to $O\left(\tau^2 p\right)$.

### 3.2 RADAGRAD: A faster approximation

From Table A.1, the expensive steps in Algorithm 1 are the update of $\mathbf{G}_t$ (line 3), the random projection (line 4) and the projection onto the approximate range of $\mathbf{G}_t$ (line 6). In the following we propose RADAGRAD, an algorithm that reduces the complexity to $O\left(\tau^2 p\right)$ by only approximately solving some of the expensive steps in ADA-LR while maintaining similar performance in practice.

To compute the approximate range $\mathbf{Q}$, we do not need to store the full matrix $\mathbf{G}_t$. Instead we only require the low dimensional matrix $\tilde{\mathbf{G}}_t = \mathbf{G}_t \mathbf{\Pi}$. This matrix can be computed iteratively by setting $\tilde{\mathbf{G}}_t \in \mathbb{R}^{p \times \tau} = \tilde{\mathbf{G}}_{t-1} + \mathbf{g}_t(\mathbf{\Pi g}_t)^\top$. This directly reduces the cost of the random projection to $O(p \log \tau)$ since we only project the vector $\mathbf{g}_t$ instead of the matrix $\mathbf{G}_t$, it also makes the update of $\tilde{\mathbf{G}}_t$ faster and saves storage.

We then project $\tilde{\mathbf{G}}_t$ on the approximate range of $\mathbf{G}_t$ and use the SVD to compute the inverse square root. Since $\mathbf{G}_t$ is symmetric its row and column space are identical so little information is lost by projecting $\tilde{\mathbf{G}}_t$ instead of $\mathbf{G}_t$ on the approximate range of $\mathbf{G}_t$.[3] The advantage is that we can now compute the SVD in $O\left(\tau^3\right)$ and the matrix-matrix product on line 6 in $O\left(\tau^2 p\right)$. See Algorithm 2 for the full procedure.

The most expensive steps are now the QR decomposition and the matrix multiplications in steps 6 and 8 (see Algorithm 2 and Table A.1). Since at each iteration we only update the matrix $\tilde{\mathbf{G}}_t$ with the rank-one matrix $\mathbf{g}_t\tilde{\mathbf{g}}_t^\top$ we can use faster rank-1 QR-updates [11] instead of recomputing the full QR decomposition. To speed up the matrix-matrix product $\tilde{\mathbf{G}}_t^\top \mathbf{Q}$ for very large problems (e.g. backpropagation in convolutional neural networks), a multithreaded BLAS implementation can be used.

### 3.3 Practical algorithms

Here we outline several simple modifications to the RADAGRAD algorithm to improve practical performance.

**Corrected update.** The random projection step only retains at most $\tau$ eigenvalues of $\mathbf{G}_t$. If the assumption of low effective rank does not hold, important information from the $p - \tau$ smallest eigenvalues might be discarded. RADAGRAD therefore makes use of the *corrected* update

$$\boldsymbol{\beta}_{t+1} = \boldsymbol{\beta}_t - \eta \mathbf{V}(\boldsymbol{\Sigma}^{1/2} + \delta \mathbf{I})^{-1}\mathbf{V}^\top \mathbf{g}_t - \boldsymbol{\gamma}_t, \quad \text{where} \qquad \boldsymbol{\gamma}_t = \eta(\mathbf{I} - \mathbf{V}\mathbf{V}^\top)\mathbf{g}_t.$$

$\boldsymbol{\gamma}_t$ is the projection of the current gradient onto the space orthogonal to the one captured by the random projection of $\mathbf{G}_t$. This ensures that important variation in the gradient which is poorly approximated by the random projection is not completely lost. Consequently, if the data has rank less than $\tau$, $\|\boldsymbol{\gamma}\| \approx 0$. This correction only requires quantities which have already been computed but greatly improves practical performance.

**Variance reduction.** Variance reduction methods based on SVRG [19] obtain lower-variance gradient estimates by means of computing a "pivot point" over larger batches of data. Recent work has shown improved theoretical and empirical convergence in non-convex problems [1] in particular in combination with ADAGRAD.

We modify RADAGRAD to use the variance reduction scheme of SVRG. The full procedure is given in Algorithm 3 in Section B. The majority of the algorithm is as RADAGRAD except for the outer loop which computes the pivot point, $\mu$ every epoch which is used to reduce the variance of the stochastic gradient (line 4). The important additional parameter is $m$, the update frequency for $\mu$. As in [1] we set this to $m = 5n$. Practically, as is standard practise we initialise RADA-VR by running ADAGRAD for several epochs.

We study the empirical behaviour of ADA-LR, RADAGRAD and its variance reduced variant in the next section.

## 4 Experiments

### 4.1 Low effective rank data

We compare the performance of our proposed algorithms against both the diagonal and full-matrix ADAGRAD variants in the idealised setting where the data is dense but has low effective rank. We generate binary classification data with $n = 1000$ and $p = 125$. The data is sampled i.i.d. from a Gaussian distribution $\mathcal{N}(\mu_c, \boldsymbol{\Sigma})$ where $\boldsymbol{\Sigma}$ has rapidly decaying eigenvalues $\lambda_j(\boldsymbol{\Sigma}) = \lambda_0 j^{-\alpha}$ with $\alpha = 1.3, \lambda_0 = 30$. Each of the two classes has a different mean, $\mu_c$.

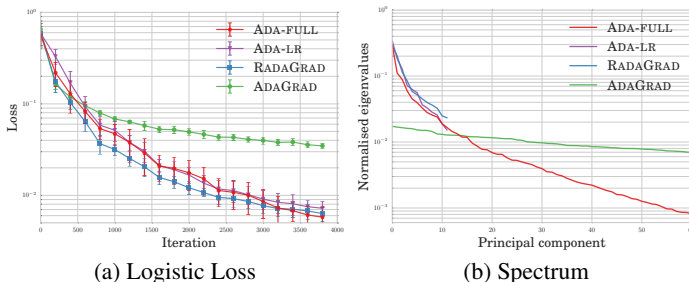

(a) Logistic Loss       (b) Spectrum

Figure 1: Comparison of: (a) loss and (b) the largest eigenvalues (normalised by their sum) of the proximal term on simulated data.

For each algorithm learning rates are tuned using cross validation. The results for 5 epochs are averaged over 5 runs with different permutations of the data set and instantiations of the random projection for ADA-LR and RADAGRAD. For the random projection we use an oversampling factor so $\boldsymbol{\Pi} \in \mathbb{R}^{(10+\tau) \times p}$ to ensure accurate recovery of the top $\tau$ singular values and then set the values of $\lambda_{[\tau:p]}$ to zero [15].

Figure 1a shows the mean loss on the training set. The performance of ADA-LR and RADAGRAD match that of ADA-FULL. On the other hand, ADAGRAD converges to the optimum much more slowly. Figure 1b shows the largest eigenvalues (normalized by their sum) of the proximal matrix for each method at the end of training. The spectrum of $\mathbf{G}_t$ decays rapidly which is matched by

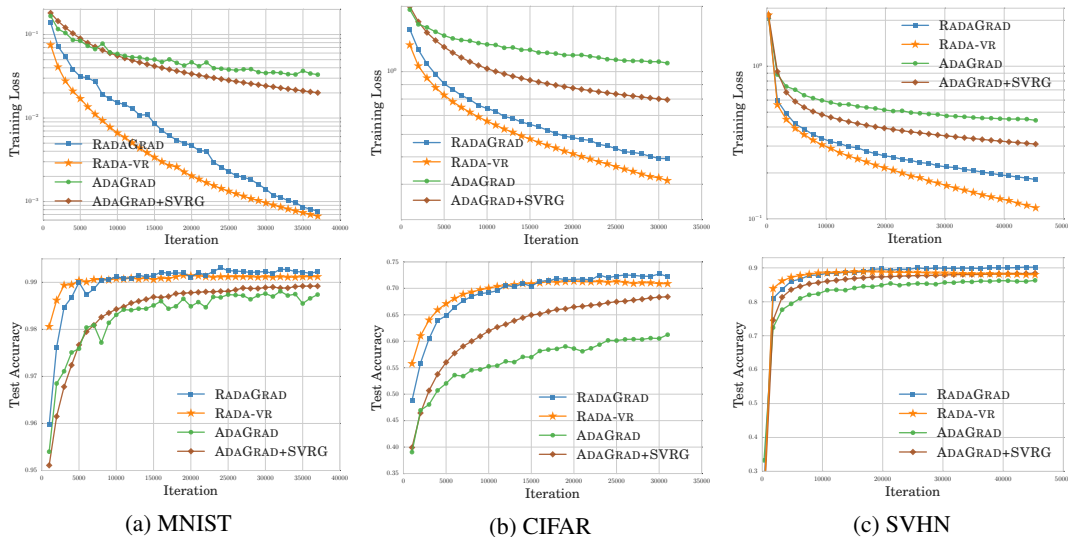

Figure 2: Comparison of training loss (top row) and test accuracy (bottom row) on (a) MNIST, (b) CIFAR and (c) SVHN.

the randomized approximation. This illustrates the dependencies between the coordinates in the gradients and suggests $\mathbf{G}_t$ can be well approximated by a low-dimensional matrix which considers these dependencies. On the other hand the spectrum of ADAGRAD (equivalent to the diagonal of $\mathbf{G}$) decays much more slowly. The learning rate, $\eta$ chosen by RADAGRAD and ADA-FULL are roughly one order of magnitude higher than for ADAGRAD.

## 4.2   Non-convex optimization in neural networks

Here we compare RADAGRAD and RADA-VR against ADAGRAD and the combination of ADAGRAD+SVRG on the task of optimizing several different neural network architectures.

**Convolutional Neural Networks.**   We used modified variants of standard convolutional network architectures for image classification on the MNIST, CIFAR-10 and SVHN datasets. These consist of three $5 \times 5$ convolutional layers generating 32 channels with ReLU non-linearities, each followed by $2 \times 2$ max-pooling. The final layer was a dense softmax layer and the objevtive was to minimize the categorical cross entropy.

We used a batch size of 8 and trained the networks without momentum or weight decay, in order to eliminate confounding factors. Instead, we used dropout regularization ($p = 0.5$) in the dense layers during training. Step sizes were determined by coarsely searching a log scale of possible values and evaluating performance on a validation set. We found RADAGRAD to have a higher impact with convolutional layers than with dense layers, due to the higher correlations between weights. Therefore, for computational reasons, RADAGRAD was only applied on the convolutional layers. The last dense classification layer was trained with ADAGRAD. In this setting ADA-FULL is computationally infeasible. The number of parameters in the convolutional layers is between 50-80k. Simply storing the full $\mathbf{G}$ matrix using double precision would require more memory than is available on top-of-the-line GPUs.

The results of our experiments can be seen in Figure 2, where we show the objective value during training and the test accuracy. We find that both RADAGRAD variants consistently outperform both ADAGRAD and the combination of ADAGRAD+SVRG on these tasks. In particular combining RADAGRAD with variance reduction results in the largest improvement for training although both RADAGRAD variants quickly converge to very similar values for test accuracy.

For all models, the learning rate selected by RADAGRAD is approximately an order of magnitude larger than the one selected by ADAGRAD. This suggests that RADAGRAD can make more aggressive steps than ADAGRAD, which results in the relative success of RADAGRAD over ADAGRAD, especially at the beginning of the experiments.

We observed that RADAGRAD performed $5$-$10\times$ slower than ADAGRAD per iteration. This can be attributed to the lack of GPU-optimized SVD and QR routines. These numbers are comparable with other similar recently proposed techniques [23]. However, due to the faster convergence we found that the overall optimization time of RADAGRAD was lower than for ADAGRAD.

**Recurrent Neural Networks.** We trained the *strongly-typed* variant of the long short-term memory network (T-LSTM, [4]) for language modelling, which consists of the following task: Given a sequence of words from an original text, predict the next word. We used pre-trained GLOVE embedding vectors [29] as input to the T-LSTM layer and a softmax over the vocabulary (10k words) as output. The loss is the mean categorical cross-entropy. The memory size of the T-LSTM units was set to 256. We trained and evaluated our network on the *Penn Treebank* dataset [25].

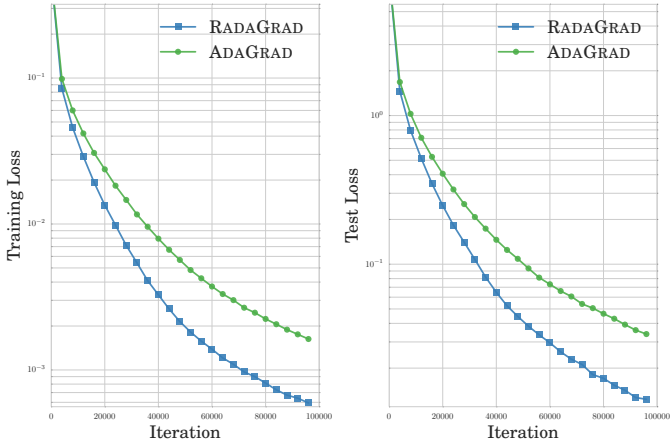

Figure 3: Comparison of training loss (left) and and test loss (right) on language modelling task with the T-LSTM.

We subsampled strings of length 20 from the dataset and asked the network to predict each word in the string, given the words up to that point. Learning rates were selected by searching over a log scale of possible values and measuring performance on a validation set.

We compared RADAGRAD with ADAGRAD without variance reduction. The results of this experiment can be seen in Figure 3. During training, we found that RADAGRAD consistently outperforms ADAGRAD: RADAGRAD is able to both quicker reduce the training loss and also reaches a smaller value ($5.62 \times 10^{-4}$ vs. $1.52 \times 10^{-3}$, a $2.7\times$ reduction in loss). Again, we found that the selected learning rate is an order of magnitude higher for RADAGRAD than for ADAGRAD. RADAGRAD is able to exploit the fact that T-LSTMs perform *type-preserving* update steps which should preserve any low-rank structure present in the weight matrices. The relative improvement of RADAGRAD over ADAGRAD in training is also reflected in the test loss ($1.15 \times 10^{-2}$ vs. $3.23 \times 10^{-2}$, a $2.8\times$ reduction).

## 5    Discussion

We have presented ADA-LR and RADAGRAD which approximate the full proximal term of ADAGRAD using fast, structured random projections. ADA-LR enjoys similar regret to ADA-FULL and both methods achieve similar empirical performance at a fraction of the computational cost. Importantly, RADAGRAD can easily be modified to make use of standard improvements such as variance reduction. Using variance reduction in combination in particular has stark benefits for non-convex optimization in convolutional and recurrent neural networks. We observe a marked improvement over widely-used techniques such as ADAGRAD and SVRG, the combination of which has recently been proven to be an excellent choice for non-convex optimization [1].

Furthermore, we tried to incorporate exponential forgetting schemes similar to RMSPROP and ADAM into the RADAGRAD framework but found that these methods degraded performance. A downside of such methods is that they require additional parameters to control the rate of forgetting.

Optimization for deep networks has understandably been a very active research area. Recent work has concentrated on either improving estimates of second order information or investigating the effect of variance reduction on the gradient estimates. It is clear from our experimental results that a thorough study of the combination provides an important avenue for further investigation, particularly where parts of the underlying model might have low effective rank.

**Acknowledgements.**    We are grateful to David Balduzzi, Christina Heinze-Deml, Martin Jaggi, Aurelien Lucchi, Nishant Mehta and Cheng Soon Ong for valuable discussions and suggestions.

## Footnotes

[2]http://www.fftw.org/

[3]This idea is similar to bilinear random projections [13].

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
