[Supplementary Material]

# Supplementary Information for Scalable Adaptive Stochastic Optimization Using Random Projections

## A    Computational Complexity

Table A.1: Comparison of computational complexity in big O notation between ADA-FULL, ADA-LR and RADAGRAD.

| Operation | Line | ADA-FULL | ADA-LR | RADAGRAD |
|---|---|---|---|---|
| $\mathbf{\Pi g}_t$ | 3 | | | $p \log \tau$ |
| $\mathbf{G}_t = \mathbf{g}_t \mathbf{g}_t^\top$ | | $p^2$ | $p^2$ | $p\tau$ |
| $\mathbf{G}_t \mathbf{\Pi}$ | 4 | | $p^2 \log \tau$ | |
| QR-decomp | 5 | | $\tau^2 p$ | $\tau^2 p$ |
| $\mathbf{Q}^\top \mathbf{G}_t$ | 6 | | $\tau p^2$ | $\tau^2 p$ |
| SVD | 7 | $p^3$ | $\tau^2 p$ | $\tau^3$ |
| $\mathbf{QW}$ | 8 | | | $\tau^2 p$ |
| $\boldsymbol{\beta}_{t+1} =$ | 10 | $p^2$ | $\tau p$ | $\tau p$ |
| Total | | $p^3$ | $\tau p^2$ | $\tau^2 p$ |

## B    RADA-VR: RADAGRAD with variance reduction.

---
**Algorithm 3** RADA-VR

---

**Input:** $\eta > 0$, $\delta \geq 0$, $\tau$, $S$ number of epochs, $m$ iterations per epoch, initial $\boldsymbol{\beta}_0^1$

1: **for** $s = 1 \dots S$ **do**
2:     $\mu = \nabla \sum_{i=1}^n f_i(\boldsymbol{\beta}_0^s)$
3:     **for** $t = 1 \dots m - 1$ **do**
4:        *Compute VR gradient:* $\mathbf{g}_t = \nabla f_t(\boldsymbol{\beta}_t^s) - \nabla f_t(\boldsymbol{\beta}_0^s) + \mu$
5:        *Project:* $\tilde{\mathbf{g}}_t = \mathbf{\Pi g}_t$
6:        $\tilde{\mathbf{G}}_t = \tilde{\mathbf{G}}_{t-1} + \mathbf{g}_t \tilde{\mathbf{g}}_t^\top$
7:        $\mathbf{Q}_t, \mathbf{R}_t \leftarrow \texttt{qr\_update}(\mathbf{Q}_{t-1}, \mathbf{R}_{t-1}, \mathbf{g}_t, \tilde{\mathbf{g}}_t)$
8:        $\mathbf{B} = \tilde{\mathbf{G}}_t^\top \mathbf{Q}$
9:        $\mathbf{U}, \mathbf{\Sigma}, \mathbf{W} = \mathbf{B}$ {SVD}
10:      $\mathbf{V} = \mathbf{W} \tilde{\mathbf{Q}}^\top$
11:      $\boldsymbol{\beta}_{t+1}^s = \boldsymbol{\beta}_t^s - \eta \mathbf{V}(\mathbf{\Sigma}^{1/2} + \delta \mathbf{I})^{-1} \mathbf{V}^\top \mathbf{g}_t - \boldsymbol{\gamma}_t$
12:     **end for**
13:     $\boldsymbol{\beta}_0^{s+1} = \boldsymbol{\beta}_{t+1}^s$
14: **end for**
**Output:** $\boldsymbol{\beta}_m^S$

---

## C    Analysis

### C.1    Regret bound for ADA-LR

The following proof is based on the proof for Theorem 7 in [9]. The key difference is that instead of having the square root and (pseudo-)inverse of the full matrix $\mathbf{G}_t$ : $\mathbf{G}_t^{1/2}$ and $\mathbf{S}_t^\dagger$ we have the approximate square root and inverse based on the randomized SVD [15]): $\tilde{\mathbf{S}}_t = (\mathbf{QQ}^\top \mathbf{G}_t)^{1/2}$ and $\tilde{\mathbf{S}}_t^\dagger = (\mathbf{QQ}^\top \mathbf{G}_t)^{-1/2}$. Essentially we use the proximal function $\psi_t = \langle \mathbf{x}, \tilde{\mathbf{S}}_t \mathbf{x} \rangle$ or $\psi_t = \langle \mathbf{x}, \tilde{\mathbf{H}}_t \mathbf{x} \rangle$ where we set $\tilde{\mathbf{H}}_t = \delta \mathbf{I} + \tilde{\mathbf{S}}_t$. Here $\mathbf{Q}$ is the approximate basis for the range of the matrix $\mathbf{G}_t$ [15].

We first state the following facts about the relationship between $\mathbf{G}$ and $\tilde{\mathbf{G}}^{-1/2}$.

**Lemma 3.** *Defining $\tilde{\mathbf{G}}^{-1/2} = (\mathbf{Q}\mathbf{Q}^\top \mathbf{G})^{-1/2}$ we have*

*(I)* $\tilde{\mathbf{G}}^{-1/2}\mathbf{G} = (\mathbf{G}^{-1}(\mathbf{Q}\mathbf{Q}^\top)\mathbf{G}^2)^{1/2}$,

*(II)* $\operatorname{tr}((\mathbf{G}^{-1}(\mathbf{Q}\mathbf{Q}^\top)\mathbf{G}^2)^{1/2}) = \operatorname{tr}(\tilde{\mathbf{G}}^{1/2})$.

We also require the following Lemma which bounds the sequence of proximal terms by the trace of the final $\tilde{\mathbf{G}}^{-1/2}$.

**Lemma 4** (Based on Lemma 10 in [9])**.**

$$\sum_{t=1}^{T} \langle \mathbf{g}_t, \tilde{\mathbf{G}}_t^{-1/2}\mathbf{g}_t \rangle \le 2\sum_{t=1}^{T} \langle \mathbf{g}_t, \tilde{\mathbf{G}}_T^{-1/2}\mathbf{g}_t \rangle = 2\operatorname{tr}(\tilde{\mathbf{G}}_T^{1/2}). \tag{3}$$

We are now ready to prove Proposition 2.

*Proof of Proposition 2.* Inspecting Lemma 6:

$$R(T) \le \frac{1}{\eta}\psi_T(\boldsymbol{\beta}^{\text{opt}}) + \frac{\eta}{2}\sum_{t=1}^{T} \|f_t'(\boldsymbol{\beta}_t)\|_{\psi_{T-1}^*}^2,$$

we first bound the term $\sum_{t=1}^{T} \|f_t'(\boldsymbol{\beta}_t)\|_{\psi_{T-1}^*}^2$.

From [9, Proof of Theorem 7] we have that the squared dual norm associated with $\psi_t$ is

$$\|\mathbf{x}\|_{\psi_t^*}^2 = \langle \mathbf{x}, (\delta\mathbf{I} + (\mathbf{Q}\mathbf{Q}^\top \mathbf{G}_t)^{1/2})^{-1}\mathbf{x} \rangle$$

and thus it is clear that $\|\mathbf{g}_t\|_{\psi_t^*}^2 \le \langle \mathbf{g}_t, (\mathbf{Q}\mathbf{Q}^\top \mathbf{G}_t)^{-1/2}\mathbf{g}_t \rangle$. Lemma 8 shows that $\|\mathbf{g}_t\|_{\psi_{t-1}^*}^2 \le \langle \mathbf{g}_t, \tilde{\mathbf{S}}_t\mathbf{g}_t \rangle$ as long as $\delta \ge \|\mathbf{g}_t\|_2$. Lemma 4 then implies that

$$\sum_{t=1}^{T} \|f_t'(\boldsymbol{\beta}_t)\|_{\psi_{T-1}^*}^2 \le 2\operatorname{tr}(\tilde{\mathbf{G}}_T^{1/2}).$$

We now bound $2\operatorname{tr}(\tilde{\mathbf{G}}_T^{1/2})$ by $2(\operatorname{tr}(\mathbf{G}_T^{1/2}) + \tau\sqrt{\epsilon})$:

$$\operatorname{tr}(\tilde{\mathbf{G}}_T^{1/2}) - \operatorname{tr}(\mathbf{G}_T^{1/2}) = \operatorname{tr}(\tilde{\mathbf{G}}_T^{1/2} - \mathbf{G}_T^{1/2}) \tag{4}$$

$$= \sum_{j=1}^{\tau}\left(\lambda_j(\tilde{\mathbf{G}}_T^{1/2}) - \lambda_j(\mathbf{G}_T^{1/2})\right) - \sum_{j=\tau+1}^{p}\lambda_j(\mathbf{G}_T^{1/2}) \tag{5}$$

$$\le \sum_{j=1}^{\tau}\left(\lambda_1(\tilde{\mathbf{G}}_T^{1/2}) - \lambda_1(\mathbf{G}_T^{1/2})\right) \tag{6}$$

since $\lambda_j(\tilde{\mathbf{G}}_T) = 0, \ \forall j > \tau$.

Now, using the reverse triangle inequality and Theorem 5 we obtain

$$\sum_{j=1}^{\tau}\left(\lambda_1(\tilde{\mathbf{G}}_T^{1/2}) - \lambda_1(\mathbf{G}_T^{1/2})\right) \le \sum_{j=1}^{\tau}\|\tilde{\mathbf{G}}_T^{1/2} - \mathbf{G}_T^{1/2}\|_2 \tag{7}$$

$$\le \sum_{j=1}^{\tau}\sqrt{\epsilon} \tag{8}$$

$$\le \tau\sqrt{\epsilon}. \tag{9}$$

It remains to show that $\psi_T(\boldsymbol{\beta}^{\mathrm{opt}})$ in Lemma 6 is bounded by $\left(\delta + \sqrt{\epsilon} + \mathrm{tr}(\mathbf{G}_T^{1/2})\right)\|\boldsymbol{\beta}^{\mathrm{opt}}\|^2$ to get the statement of Theorem 2:

$$\psi_T(\boldsymbol{\beta}^{\mathrm{opt}}) = \langle \boldsymbol{\beta}^{\mathrm{opt}}, \delta\mathbf{I} + (\mathbf{Q}\mathbf{Q}^\top\mathbf{G}_T)^{1/2}\boldsymbol{\beta}^{\mathrm{opt}}\rangle$$
$$\leq \|\boldsymbol{\beta}^{\mathrm{opt}}\|^2\|(\mathbf{Q}\mathbf{Q}^\top\mathbf{G}_T)^{1/2}\|_2 + \delta\|\boldsymbol{\beta}^{\mathrm{opt}}\|^2$$
$$\leq \|\boldsymbol{\beta}^{\mathrm{opt}}\|^2\left(\sqrt{\epsilon} + \|\mathbf{G}_T^{1/2}\|\right) + \delta\|\boldsymbol{\beta}^{\mathrm{opt}}\|^2$$
$$\leq \|\boldsymbol{\beta}^{\mathrm{opt}}\|^2\left(\sqrt{\epsilon} + \mathrm{tr}(\mathbf{G}_T^{1/2})\right) + \delta\|\boldsymbol{\beta}^{\mathrm{opt}}\|^2$$

where we again use the reverse triangle inequality and Theorem 5 as above.

Finally, plugging this into the statement of Lemma 6 and setting $\eta = \|\boldsymbol{\beta}^{\mathrm{opt}}\|_2$ (as in Corollary 11 in [9]) we get the expression for the regret of ADA-LR as stated in Theorem 2. $\qquad\square$

### C.2 Proofs of supporting results

*Proof of Lemma 3.* By direct computation we have for (I)

$$\tilde{\mathbf{G}}^{-1/2}\mathbf{G} = (\mathbf{Q}\mathbf{Q}^\top\mathbf{G})^{-1/2}\mathbf{G}$$
$$= ((\mathbf{Q}\mathbf{Q}^\top\mathbf{G})^{-1}\mathbf{G}^2)^{1/2}$$
$$= (\mathbf{G}^{-1}(\mathbf{Q}\mathbf{Q}^\top)^{-1}\mathbf{G}^2)^{1/2}$$
$$= (\mathbf{G}^{-1}(\mathbf{Q}\mathbf{Q}^\top)\mathbf{G}^2)^{1/2}.$$

and for (II)

$$\mathrm{tr}((\mathbf{G}^{-1}(\mathbf{Q}\mathbf{Q}^\top)\mathbf{G}^2)^{1/2}) = \mathrm{tr}((\mathbf{Q}^\top\mathbf{G}\mathbf{Q})^{1/2})$$
$$= \mathrm{tr}((\mathbf{Q}\mathbf{Q}^\top\mathbf{G})^{1/2})$$
$$= \mathrm{tr}(\tilde{\mathbf{G}}^{1/2}).$$

$\qquad\square$

*Proof of Lemma 4.* We set up the following proof by induction. In the base case:

$$\langle \mathbf{g}_1, \tilde{\mathbf{G}}_1^{-1/2}, \mathbf{g}_1\rangle = \mathrm{tr}(\tilde{\mathbf{G}}_1^{-1/2}\mathbf{g}_1\mathbf{g}_1^\top) = \mathrm{tr}(\tilde{\mathbf{G}}_1^{1/2}) \leq 2\mathrm{tr}(\tilde{\mathbf{G}}_1^{1/2}),$$

where we have used (II).

Now, assuming that the lemma is true for $T-1$, we get:

$$\sum_{t=1}^T \langle g_t, \tilde{\mathbf{G}}_t^{-1/2}, g_t\rangle \leq 2\sum_{t=1}^T \langle \mathbf{g}_t, \tilde{\mathbf{G}}_{T-1}^{-1/2}\mathbf{g}_t\rangle + \langle \mathbf{g}_T, \tilde{\mathbf{G}}_T^{-1/2}\mathbf{g}_T\rangle.$$

Now using that $\tilde{\mathbf{G}}_{T-1}^{-1/2}$ does not depend on $t$ and (II):

$$\sum_{t=1}^{T-1} \langle g_t, \tilde{\mathbf{G}}_{T-1}^{-1/2}g_t\rangle = \mathrm{tr}(\tilde{\mathbf{G}}_{T-1}^{-1/2}\mathbf{G}_{T-1}) = \mathrm{tr}(\tilde{\mathbf{G}}_{T-1}^{1/2}).$$

Therefore we get

$$\sum_{t=1}^T \langle g_t, \tilde{\mathbf{G}}_t^{-1/2}g_t\rangle \leq 2\mathrm{tr}(\tilde{\mathbf{G}}_{T-1}^{1/2}) + \langle \mathbf{g}_T, \tilde{\mathbf{G}}_T^{-1/2}\mathbf{g}_T\rangle. \tag{10}$$

We can rewrite

$$\mathrm{tr}(\tilde{\mathbf{G}}_{T-1}^{1/2}) = \mathrm{tr}\left(\left(\mathbf{Q}_{T-1}\mathbf{Q}_{T-1}^\top\mathbf{G}_T - \mathbf{Q}_{T-1}\mathbf{Q}_{T-1}^\top\mathbf{g}_T\mathbf{g}_T^\top\right)^{1/2}\right) \tag{11}$$

Now since $\mathrm{range}(\mathbf{Q}_{T-1}) \subset \mathrm{range}(\mathbf{Q}_T)$ and Proposition 8.5 in [15] we can use Lemma 7 with $\nu = 1$ and $\mathbf{g} = \mathbf{g}_t$ to obtain:

$$2\mathrm{tr}(\tilde{\mathbf{G}}_{T-1}^{1/2}) + \langle \mathbf{g}_T, \tilde{\mathbf{G}}_T^{-1/2}, \mathbf{g}_T\rangle \leq 2\mathrm{tr}(\tilde{\mathbf{G}}_T^{1/2}) \tag{12}$$

$\qquad\square$

# D   Supporting Results

**Theorem 5** (SRFT approximation error (Theorem 11.2 in [15])). *Defining $\epsilon = \sqrt{1 + 7p/\tau} \cdot \sigma_{k+1}$ the following holds with failure probability at most $O\left(k^{-1}\right)$*

$$\left\|\mathbf{G}_t - \mathbf{Q}\mathbf{Q}^\top \mathbf{G}_t\right\|_2 \leq \epsilon, \tag{13}$$

*where $\sigma_{k+1}$ is the kth largest singular value of $\mathbf{G}_t$, and $4\left[\sqrt{k} + \sqrt{8\log(kn)}\right]^2 \leq \tau \leq p$.*

**Lemma 6** (Proposition 2 from [9]).

$$R(T) := \sum_{t=1}^{T} f_t(\boldsymbol{\beta}_t) + \varphi(\boldsymbol{\beta}_t) - f_t(\boldsymbol{\beta}^{opt}) - \varphi(\boldsymbol{\beta}^{opt}) \leq \frac{1}{\eta}\psi_T(\boldsymbol{\beta}^{opt}) + \frac{\eta}{2}\sum_{t=1}^{T}\|f_t'(\boldsymbol{\beta}_t)\|_{\psi_{T-1}^*}^2$$

**Lemma 7** (Lemma 8 from [9]). *Let $\mathbf{B} \succeq 0$. For any $\nu$ such that $\mathbf{B} - \nu\mathbf{g}\mathbf{g}^\top \succeq 0$ the following holds*

$$2\mathrm{tr}((\mathbf{B} - \nu\mathbf{g}\mathbf{g}^\top)^{1/2}) \leq 2\mathrm{tr}(\mathbf{B}^{1/2}) - \nu\mathrm{tr}(\mathbf{B}^{1/2}\mathbf{g}\mathbf{g}^\top)$$

**Lemma 8** (Lemma 9 from [9]). *Let $\delta \geq \|\mathbf{g}\|_2$ and $\mathbf{A} \succeq 0$, then*

$$\langle \mathbf{g}, (\delta\mathbf{I} + \mathbf{A}^{1/2})^{-1}\mathbf{g}\rangle \leq \langle \mathbf{g}, ((\mathbf{A} + \mathbf{g}\mathbf{g}^\top)^\dagger)^{1/2}\mathbf{g}\rangle$$