[Reviews · NeurIPS 2016]

Reviewer 1

Summary

This paper considers faster stochastic gradient algorithms for possibly non-convex functions. In particular this paper considers various schemes to improve upon AdaGrad and its full matrix variant AdaFull, popular iterative methods for solving such problems over large data sets. More precisely, this paper provides an algorithm called Ada-LR that uses sketching techniques to obtain convergence guarantees similar to AdaFull without paying for the iteration costs. The paper provides a proof of the regret obtained by this algorithm to demonstrate its efficacy over AdaGrad and then discuss various improvements to decrease the iteration costs, make the algorithm more stable, and perform variance reduction. Finally, the paper provides an empirical evaluation of these methods on various problems suggesting how they may improve the performance of deep network training in some regimes.

Qualitative Assessment

Using fast randomized numerical linear algebra techniques to provably and practically improve the performance of AdaGrad and AdaFull is a wonderful idea and this paper initiates what could be a very interesting line of work in this area. However, the particular algorithm and modifications they propose are quite natural and consequently it is difficult to pinpoint where the novelty in the approach is. More troubling, it seems like details of the algorithms performance may be hidden in this size of the sketch matrix Pi and how frequently it is computed. It would be nice if the main theorem (Proposition 2) made clear the end-to-end cost of the algorithm and what it depends on. Also, note that properties of a sketch matrix Pi may not hold under adaptive queries. Note the vectors g_t+1 depend on beta_t+1 which in turn dependent on Pi. Given this, how is it ensured that Pi has the right properties in such a scenario if it is not recomputed? I believe Theorem 11.2 in [14] only holds if the matrix sketched is fixed apriori. Nevertheless, the empirical success of the algorithm, the cleanliness of the paper, and its practical utility may be sufficient to counteract the above concerns. Detailed Comments • Line 33-35: just in certain cases, right? • Line 53: I would be a little more specific with what mean by “AdaGrad style algorithms” • Line 67: Why is this impractical if it was just a low space single pass? • Line 70 – 81: Maybe subspace embeddings should be included in this discussion? • Line 93: Why was G_t-1 + g_tg_t^T written instead of just G_t ? • Line 124: What is M? • Section 3.2 and 3.3 are there any theoretical guarantees for these algorithms?

Confidence in this Review

2-Confident (read it all; understood it all reasonably well)


Reviewer 2

Summary

This paper proposes an extension is Adagrad - a method for adaptively adjusting the learning rates per individual feature, to a full second order version, called Ada-Full, while assuming that the resulting second order matrix is low rank and hence the system of linear equations that needs to be solved is solved via random projections. This results in a significant improvement in per iteration complexity while maintaining desired improvement with high probability on each step. Additionally, the authors propose further improvement by employing inexact computations in each step. This allows for even cheaper iteration complexity, but there is no theory provided for this improvement. Yet it seems to work well in practice.

Qualitative Assessment

I find the paper very clearly written, with well defined goals and contributions. I see the use of randomized linear algebra as very significant in the future development of optimization algorithms. My only puzzlement as to why the authors chose the analysis of the online (regret) setting, while their implementation seems to target regular optimization setting for ML. I assume the choice is due to availability of theoretical results for regret, rather than regular objective convergence rates. The difference is not very significant, of course.

Confidence in this Review

2-Confident (read it all; understood it all reasonably well)


Reviewer 3

Summary

This paper proposed low-rank appromixation of a core matrix. A randomized version is also provided. Regret bound is also given and experiments showed advantages in speed.

Qualitative Assessment

Despite its usefulness, the proposed method is a bit trivial. Low-rank approximation and using randomized SVD for low-rank approximation are well known techniques. Besides, I have two minor comments: 1. Why the randomized algorithm can improve performance (lower objective value and higher recognition rate)? See Fig. 1(a) and Fig. 2(a)-(c). It anyway solves the same mathematical model. 2. I regret that the scales of experiments are all too small. ********************* Comments after reading authors' feedback: It is not me only who thought that the paper is trivial. Reviewer_5 had the same opinion as mine. Maybe it is because we are more familiar with the low-rank techniques and randomized algorithms. It is just a standard implementation of rank-one update (which can be as early as BFGS), where randomized SVD may come in. The technique is definitely useful, but may not be good enough for NIPS by my personal view. As for the author responses to my comments, I don't think they are satisfctory. For randomized algorithms, although in each iteration the computation is saved, they nonetheless produce less accurate solutions, thus normally requiring more iterations to achieve the same precision. However, I did not see such an effect in their experiments. On the contrary, the proposed randomized algorithm is even more accurate than ADA-FULL, the one using the full matrix! If the x-axis in Figures 1&2 are running time, I would believe the results. But they are iteration numbers. So I don't think that the experimental results are trustworthy. The authors wrote in the feedback that "RadaGrad improves on AdaGrad because it exploits effective low-rank structure of the matrix G. AdaGrad only stores a diagonal approximation" but this did not answer why RadaGrad is better than ADA-FULL, as using the full matrix should be the most accurate. As for the scale of problems, are 10K parameters large? It only corresponds to a 100x100 matrix, while I typically computed with 3000x3000 matrices. Only with large matrices can one see how inaccurate low-rank approximation could be! The number, 100K, of samples should not be an issue: they could be read into memory by batches.

Confidence in this Review

2-Confident (read it all; understood it all reasonably well)


Reviewer 4

Summary

This paper formally proposes two algorithms: ADA-LR and RADAGRAD, which are computationally efficient approximations of ADAGRAD when using a full matrix approximation for the proximal term. The authors use ADA-FULL for this full-matrix method and use ADAGRAD to refer to the most common variant where one uses a diagonal matrix approximation. Though ADAGRAD performs well in practice, the authors argue that when data has low effective rank, the regret of ADAGRAD is substantially worse than ADA-FULL, and the problem is exacerbated in high dimensions (e.g., the combined weight vector for deep neural networks). This therefore motivates the use of their two algorithms to approximate ADA-FULL. ADA-LR is an approximation based on random projections, and RADAGRAD takes this a step further by using more randomized methods. The authors also identify additional improvements to RADAGRAD for practical performance benefits. Their experimental results on image recognition (using convolutional neural networks) and language modeling (using long short term memory networks) show that the performance of their algorithms is comparable to or superior than the state of the art.

Qualitative Assessment

I am generally impressed with this paper. The introduction gives convincing arguments for the need to modify ADAGRAD/ADA-FULL in the prevalence of low effective rank data. Then the paper introduces and describes the new algorithms as a series of matrix operations to approximate ADA-FULL. Then it provides some theoretical results (deferring proofs to the appendix) and presents three sets of experiments. This is a logical progression of the material. In the related work section, you might wish to remove the second paragraph in favor of other references that argue about the prevalence of low effective rank data. The second paragraph, while certainly conceptually related to the paper, doesn't have anything saying its explicit relation. For instance, having something like "variance reduced SGD algorithms have vastly superior performance BUT perform extremely poorly on low effective rank data" would be helpful to understand its context. I suggest this because the paper's contribution is only going to be as good as the prevalence of low effective rank data in the real world, and this is the recurring theme that must be emphasized. The algorithms appear useful and insightful, and involve "just" a series of matrix operations simple enough so that an alert reader should be able to code it up. It seems a little awkward to include ADA-VR when, as far as I can tell, there's virtually no reason to use it over RADAGRAD, but it makes the paper's story proceed nicely so I guess it's OK. The table in Appendix A is enormously helpful. Thank you for including this. The experiments the authors chose to run seem reasonable to me given the research goals (and the 9-page space constraint). The first experiment is logical to include since it provides the best-case scenario for the algortihms, and they certainly perform well. The second experiment presented in Figure 2 is, to me, the most interesting one of the paper. My main concern there pertains to your comment about using RADAGRAD only for the convolutional layers. It seems like a fairer comparision would be to use RADAGRAD for everything or ADAGRAD for everything? I think more discussion about this is worthier than the earlier discussion about the implementation; there's little reason to talk about the code beyond that it was implemented in Lasagne/Theano/Python. I am also intrigued by the third experiment due to the growing importance of LSTMs nowadays. However, the performance benefits over ADAGRAD+SVRG seem extremely minor, even when taking into account the log scale. Is there a way to explain intuitively what this extra benefit means? You might explain this and delete the second paragraph in Section 5, which doesn't seem to make sense for the paper (the forgetting schemes were never discussed earlier). Despite some nitpicks over the experiments, as mentioned earlier I have a positive impression on this paper, and lean towards its acceptance to NIPS. It introduces an important extension to the widely-used ADAGRAD function and argues that these algorithms are needed when we have low effective rank data. While it is not as groundbreaking as the original ADAGRAD paper, I believe the research contribution is substantial enough to warrant consideration to NIPS. Minor comments: In line 108, you might wish to say "borrowing the $g_{1:T,j}$ notation" or something like that, because the notation is never explicitly defined in this paper, but is used later (e.g., in Equation 2). In lines 124 through 128 and including Equation 2, I am unable to derive the first part of Equation 2. I am also unsure why you assume $g_t \propto Mx_t$, because M is later revealed to be a scalar, so why the need for the proportional symbol? That constant can be absorbed into M. I once thought it should be something like $\|g_t\|_2 \le M\|x_t\|_2$ but perhaps the authors could clarify. In line 145, did you mean to say "applicable in this low-dimensional *space*"? In line 192, "it's" should be "its". In Figure 1, please be consistent and use ADA-FULL instead of ADA-F, since the former is used throughout the paper's text. In Figure 1(b), you might consider using the same number of principal components for RADAGRAD and ADA-LR. I think one has 21 and another has 28, and I believe these values come from the top $\tau$ singular values, so you might want to make them equal, or otherwise explain your choice for $\tau$. I'm not sure why there's a difference right now. I'm assuming $\tau$ is just chosen heuristically? In line 268, "A the end" should be "At the end". In line 397 (in Appendix), the title of C.1 is misleading, it should be the regret bound for ADA-LR, not RADAGRAD.

Confidence in this Review

2-Confident (read it all; understood it all reasonably well)


Reviewer 5

Summary

This paper presents two new approximations of the proximal term of ADA-FULL that are different than the one used by ADAGRAD. The first one ADA-LR use random projection to reduce the complexity of ADA-FULL while keeping a similar regret bound with high probability. The second, RADAGRAD, reduces even further this complexity. The experimental section compares the new approaches with ADAGRAD on synthetic low rank datas. It also compares RADAGRAD/RADA-VR and ADAGRAD on the optimization of CNN and RNN.

Qualitative Assessment

- Some notation problems that make the paper hard to understand properly: - Left part of equation (1) has the norms properly specified but everywhere else the norms are not specified. - The notations in 3.1 does not match the ones in Algorithm 1. Especially most of the "tilde" notations are not defined properly - The title of C.1 does not match with what is present inside and what is stated in the paper. Is the bound for RADAGRAD the same as ADA-LR? what about the version with the corrected update? - The corrected update of RADAGRAD presented in 3.3 is not present in Algorithm 2 that presents RADAGRAD. - The experimental section convince me that RADAGRAD converges faster in the number of iteration compared to ADAGRAD. Though I would be interested in a comparison with respect to the time to see if the proposed algorithm has a practical run-time.

Confidence in this Review

2-Confident (read it all; understood it all reasonably well)


Reviewer 6

Summary

The paper is attempting to approximate full matrix AdaGrad by using dimensionality reduction by random projections. They are proposing two methods that they claim are close to full matrix AdaGrad in performance but with a fraction of the computation cost. Particularly, the claim that these methods are better than diagonal AdaGrad is of worth. 3 methods are proposed and analyzed. ADA-LR, RadaGrad and a version of RadaGrad with reduced variance. Comparison is done with AdaGrad and a variance reducing method (SVRG) on 3 tasks, one of which is synthetic.

Qualitative Assessment

The authors tackle an interesting and relevant problem. Their approach to the problem is also satisfying. Their analysis of the reason and applicability of their solution, however, is not very convincing. The experiments do not showcase the qualities of their method with respect to other methods properly. It would have been interesting to see the final accuracies for the different methods on all the tasks. If the final accuracies end up being comparable, that is also a relevant factor, as the 5-10x slowdown that this method causes might make both methods much more comparable. The last experiment (with LSTMs for language modeling), does not compare all the methods. The reason for this is not given. The graphs shown might also be potentially misleading. Given that these methods are 5-10x slower, a competetive time-scale analysis would also have been very illuminating. Since the method is not convertible to adaptive gradient methods, as the authors pointed out, it is relevant to compare it to these methods. This has not been done. There is also a typo in section 3.1. When showing how to calculate the pseudo-inverse of the proximal term H_t, the authors refer to matrix S, which is not referred to in the next up to or after that point. Based on algorithm 1, I believe this refers to \Sigma that is written as S in step 7.

Confidence in this Review

2-Confident (read it all; understood it all reasonably well)